# Effects of Dickkopf-1 (DKK-1) on Prostate Cancer Growth and Bone Metastasis

**DOI:** 10.3390/cells12232695

**Published:** 2023-11-24

**Authors:** Shiyu Yuan, Nathan K. Hoggard, Noriko Kantake, Blake E. Hildreth, Thomas J. Rosol

**Affiliations:** 1Department of Biological Sciences, The Molecular and Cellular Biology Program, College of Arts and Sciences, Ohio University, Athens, OH 45701, USA; sy920817@ohio.edu; 2Department of Biomedical Sciences, Heritage College of Osteopathic Medicine, Ohio University, Athens, OH 45701, USA; nh381421@ohio.edu (N.K.H.); kantake@ohio.edu (N.K.); 3Department of Pathology, School of Medicine, University of Alabama at Birmingham, Birmingham, AL 35294, USA; bhildreth@uabmc.edu

**Keywords:** prostate cancer, DKK-1, osteoblastic bone metastasis, canine, dog

## Abstract

Osteoblastic bone metastases are commonly detected in patients with advanced prostate cancer (PCa) and are associated with an increased mortality rate. Dickkopf-1 (DKK-1) antagonizes canonical WNT/β-catenin signaling and plays a complex role in bone metastases. We explored the function of cancer cell-specific DKK-1 in PCa growth, metastasis, and cancer–bone interactions using the osteoblastic canine PCa cell line, Probasco. Probasco or Probasco + DKK-1 (cells transduced with human *DKK-1*) were injected into the tibia or left cardiac ventricle of athymic nude mice. Bone metastases were detected by bioluminescent imaging in vivo and evaluated by micro-computed tomography and histopathology. Cancer cell proliferation, migration, gene/protein expression, and their impact on primary murine osteoblasts and osteoclasts, were evaluated in vitro. DKK-1 increased cancer growth and stimulated cell migration independent of canonical WNT signaling. Enhanced cancer progression by DKK-1 was associated with increased cell proliferation, up-regulation of NF-kB/p65 signaling, inhibition of caspase-dependent apoptosis by down-regulation of non-canonical WNT/JNK signaling, and increased expression of epithelial-to-mesenchymal transition genes. In addition, DKK-1 attenuated the osteoblastic activity of Probasco cells, and bone metastases had decreased cancer-induced intramedullary woven bone formation. Decreased bone formation might be due to the inhibition of osteoblast differentiation and stimulation of osteoclast activity through a decrease in the OPG/RANKL ratio in the bone microenvironment. The present study indicated that the cancer-promoting role of DKK-1 in PCa bone metastases was associated with increased growth of bone metastases, reduced bone induction, and altered signaling through the canonical WNT-independent pathway. DKK-1 could be a promising therapeutic target for PCa.

## 1. Introduction

Prostate cancer (PCa) is a common malignancy in men and frequently develops osteoblastic bone metastases in the late stages of the disease. Cancer-induced bone formation is due to the disruption of the balance between bone-forming osteoblasts and bone-resorbing osteoclasts [1]. Several mechanisms have been proposed to contribute to osteoblastic bone metastases, including osteoblast factors secreted by PCa (WNTs [2], BMPs [3], endothelin-1 [4]), extracellular vesicles [5] generated from PCa, and osteomimicry [6] properties acquired by PCa.

Dickkopf-1 (DKK-1), a secretory antagonist of the Wingless/integrated-1 (WNT) signaling pathway, has functionality in both bone homeostasis and cancer progression. In bones, DKK-1 inhibits the canonical WNT (β-catenin-dependent) pathway by competing with the co-receptor LRP5/6 for WNT ligands [7]. DKK-1 inhibits the commitment of the osteoblast lineage in bone marrow-derived mesenchymal stem cells (MSC) and promotes MSC differentiation into adipocytes, resulting in reduced bone formation. In addition, DKK-1 can enhance RANKL levels in the bone microenvironment, and the resultant decrease in the OPG/RANKL ratio stimulates osteoclast-driven bone resorption [8]. In cancer, the role of DKK-1 is complex, with a growing body of evidence showing that it can either inhibit cancer metastases in ovarian cancer [9] and colorectal cancer [10] or promote cancer progression in osteosarcoma [11], breast cancer [12], and prostate cancer [13]. Mechanistically, despite exerting functions in canonical WNT signaling, DKK-1 is more complex since it may also function through a non-canonical (β-catenin-independent) WNT signaling pathway. It has been reported that DKK-1 induces cancer growth through the activation of the non-canonical WNT/PCP pathway through the activation of JNK proteins [14]. In addition, the identification of the novel DKK-1 receptor, cytoskeleton-associated protein 4 (CKAP4), showed that DKK-1 functions in the regulation of P13K/AKT or NF-kB signaling, leading to tumorigenesis [15].

JNK (C-Jun N-terminal kinase), a member of mitogen-activated protein kinases (MAPKs), is a hub protein that modulates apoptosis, proliferation, and inflammation in cancer [16]. JNK has a tumorigenic effect in PCa, and the JNK inhibitor SP600125 significantly inhibited the proliferation of human osteolytic PC3 cells [17] and canine osteoblastic/osteolytic Ace-1 cells [18]. Nevertheless, JNK has also been reported to function as a tumor suppressor by inducing apoptosis and regulating anti-proliferative activity in human osteoblastic LNCaP and C4-2 cells [19]. These opposing roles of JNK in PCa imply that the function of JNK in cancer progression might depend on the phenotype of bone metastases induced by PCa.

Aberrant nuclear factor kappa-B (NF-kB) activation is common in cancers and has been implicated in the stimulation of cell proliferation, inhibition of apoptosis, and the induction of cell transformation [20]. Constitutive activation of NF-kB/p65 has been shown in human prostate adenocarcinoma and correlates with disease progression [21]. JNK and NF-kB function reciprocally to control cell survival and death. Increased cell proliferation from upregulation of NF-kB was partly due to the suppression of JNK, and the induction of apoptosis by downregulation of NF-kB was related to the sustained activation of JNK signaling [22].

This study aimed to explore the molecular role of DKK-1 in PCa progression and cancer–bone interactions using the canine osteoblastic Probasco PCa cell line. The dog is the only species other than man to spontaneously develop PCa and subsequent osteoblastic bone metastases, making it valuable as a research model [23]. Previously, we showed that Probasco cells directly induced new bone formation in mouse calvaria in vitro, and DKK-1 was able to inhibit mineralization [24]. Here, we report that DKK-1 has both autocrine and paracrine effects in the bone metastasis milieu, increasing tumor growth and decreasing the osteoblastic activity of PCa.

## 2. Materials and Methods

### 2.1. Cell Culture and Conditioned Medium (CM)

The canine Probasco cell line (derived from a dog with a spontaneous prostate carcinoma by the Rosol laboratory) [25], Probasco + DKK-1 cells (Probasco cells transduced with human *DKK-1* cDNA) [24], and Probasco-luc cells and Probasco + DKK-1-luc cells (cells transduced with the retroviral yellow fluorescent protein–luciferase dual reporter gene) were cultured in DMEM/F12/GlutaMaxTM (Gibco, Grand Island, NY, USA) containing 10% fetal bovine serum (FBS) (Gibco), 1% penicillin/streptomycin (Life Technologies, Carlsbad, CA, USA), and 100 μg/mL Normocin (Invivogen, San Diego, CA, USA). All cells were serially passage using TrypLETM (Gibco) and maintained at 37 °C and 5% CO_2_ in a humidified atmosphere. A conditioned medium (CM) for treating murine bone cells was prepared by incubating near-confluent Probasco or Probasco + DKK-1 cells in a T75 flask for 24 h with serum-free culture media. CM was stored in aliquots at −80 °C until used.

### 2.2. In Vitro Proliferation and Migration

Proliferation—Probasco (passage p22) and Probasco + DKK-1 cells (passage p23) were plated in a six-well culture plate in triplicate at a density of 200,000 cells per well. The cells were collected by trypsinization at 24, 48, and 72 h after the initial plating and counted with a TC20 automated cell counter (Bio-Rad, Hercules, CA, USA) using trypan-blue dye (Gibco) exclusion to differentiate between live and dead cells.

Migration—Probasco and Probasco + DKK-1 cells were seeded in a 2-well culture-insert (ibidi culture-insert 2 well, ibidi GmbH, Martinsried, Germany) at a density of 1 × 10^5^ cells/mL. After allowing the cells to reach confluence, the culture-insert was removed to create a 500 µm gap in the middle, and cells were washed with PBS to remove non-adherent cells. Cells were cultured in DMEM/F12 containing 1% FBS and photographed at 0, 6, 12, and 24 h to monitor cell migration. Images were analyzed using ImageJ (NIH, version 2.14.0, Bethesda, MD, USA) to show the rate of cell migration (the percentage of open gap area = gap areas at different time points/the gap area recorded at 0 h).

### 2.3. Murine Primary Bone Cell Differentiation Assay

Osteoblasts—Primary osteoblasts were isolated from calvaria from neonatal C57BL/6 mouse pups and cultured in α-MEM/GlutaMax^TM^ medium (Gibco) supplemented with 10% FBS as previously described [26]. When cells reached confluence, osteoblasts (1 × 10^5^ cell/well) were plated in 12-well plates and treated with 50% conditioned medium collected from Probasco or Probasco + DKK-1 cells, then mineralization was induced using 250 µM ascorbic acid and 10 mM β-glycerophosphate (Thermo Fisher Scientific, Waltham, MA, USA) for 7 days for qRT-PCR analysis and 14 days for von Kossa staining. Von Kossa staining for mineralization was performed by fixing cells in 95% ethanol and staining with silver nitrate and hydroquinone (Thermo Fisher Scientific) under UV light. Stained wells were imaged using a digital camera (Nikon D3500), and the percentage of the mineralized area was calculated in the central region (2 mm away from the edge) of each well using ImageJ (NIH, version 2.14.0).

Osteoclasts—Bone marrow cells were extracted from long bones of 3–4-week-old C57BL/6 mice (The Jackson Laboratory, Bar Harbor, ME, USA) and plated in α-MEM/GlutaMax^TM^ medium supplemented with 10% FBS and 50 ng/mL macrophage colony-stimulating factor CSF-1 (Peprotech, Cranbury, NJ, USA) on non-tissue culture-treated plates. After three days, cells (2 × 10^5^ cell/well) were plated in 6-well tissue culture plates treated with 20% conditioned medium from Probasco or Probasco + DKK-1 cells and then differentiated into osteoclasts using 50 ng/mL CSF-1 and 100 ng/mL RANKL (Peprotech) for 3 days for qRT-PCR and 5 days for TRAP staining. Tartrate-resistant acid phosphatase (TRAP) staining was performed according to the manufacturer’s protocol (Acid Phosphatase, Leukocyte (TRAP) Kit, Sigma–Aldrich, St Louis, MO, USA). Images of stained cells were taken using the EVOS M7000 Imaging System (Thermo Fisher). Osteoclasts that had more than three nuclei were considered TRAP-positive multinucleated cells. The number of osteoclasts was quantified using ImageJ (NIH, version 2.14.0).

### 2.4. RNA Extraction and qRT-PCR

RNA was isolated from canine PCa cells, murine osteoblasts, and osteoclasts using a PureLink^TM^ RNA mini kit (Invitrogen). Superscript II First Strand cDNA synthesis kit (Invitrogen) was used to reverse transcribe total RNA (0.5 µg) into cDNA. The qRT-PCR was performed using SYBR green supermixes (Bio-Rad) as described by the manufacturer. For canine cDNA samples, the expression of the reference gene (*GAPDH*), EMT-related genes (*CDH1*, *SNAIL*, *SLUG*, and *TWIST1*), transduced human *DKK-1* gene, and NF-kB downstream target genes (*COX2* and *VEGFA*) were detected. For murine cDNA samples, the expression of the reference gene (*Ubc*), osteoblast-related genes (*Rankl* and *Opg*), and osteoclast-related genes (*Mmp9* and *Ctsk*) were detected. The primers are listed in Table 1. The thermal cycle conditions for target genes were as follows: 95 °C for 3 min followed by 39 cycles of 95 °C for 10 s, 55 °C for 30 s, 75 °C for 30 s; then 95 °C for 1 min and 55 °C for 1 min. The expression of targeted genes was normalized by reference genes and quantified using the 2^−ΔΔCt^ method.

### 2.5. Intratibial and Intracardiac Injections of Cancer Cells into Aythmic Nude Mice

All experimental animal procedures were approved by the Ohio University Institutional Laboratory Animal Care and Use Committee (IACUC protocol number: 20-H-019).

Intratibial (IT)—5-week-old athymic nude mice (*n* = 10 per group) (The Jackson Laboratory) were anesthetized with isoflurane (2.5%) and maintained in a supine position for intratibial (IT) injection. The right rear limb was cleaned with 70% ethanol. The leg was held and the knee joint was bent, resulting in a 90° angle between the femur and the tibia. Probasco-luc or Probasco + DKK-1-luc cells (200,000 cells suspended in 10 µL sterilized DPBS (Gibco)) were loaded into a Hamilton syringe with a 27-gauge needle. The needle was inserted through the patellar ligament and articular cartilage into the marrow of the tibia’s metaphysis. The growth of the IT-injected tumor was monitored every week using bioluminescent imaging. Tumors were allowed to grow in vivo for 4 weeks, and injected mice were euthanized at that time.

Intracardiac (IC)—5-week-old athymic nude mice (*n* = 7 per group) were anesthetized using 2.5% isoflurane with an oxygen flow rate of 2 L/min. The mice were placed on their backs with chests facing upwards, and their front limbs were secured perpendicular to the thorax with surgical tape. The ventral thorax was cleaned with 70% ethanol. A 1 mL syringe with 0.1 mL of sterile DPBS containing 300,000 Probasco-luc or Probasco + DKK-1-luc cells and 0.1 mL of air between the plunger and the cell suspension was prepared. The syringe, attached to a 27-gauge needle, was vertically inserted into the left ventricle of the heart through the third intercostal space (in between the third and fourth ribs on the left of the mouse, approximately 1 mm to the left of the sternum). Once a jet of bright red pulsating blood was seen in the needle hub, the cell suspension was injected slowly over 30 s. Bioluminescent imaging was performed immediately after injection (within 5–10 min) to confirm successful IC injection. Weekly imaging was conducted to monitor for metastasis. Tumors were allowed to grow for 5 weeks, and injected mice were euthanized at that time.

### 2.6. Bioluminescent Imaging

D-Luciferin (15 mg/mL dissolved in sterile DBPS) (PerkinElmer, Waltham, MA, USA) was injected intraperitoneally in each mouse (from IT- or IC-injected mice, 10 µL/g of body weight) before anesthesia using a 1 mL insulin syringe. Mice were anesthetized with 2.5% isoflurane with an oxygen flow rate of 2 L/min for bioluminescent imaging. The IVIS 100 (Caliper Life Sciences, Hopkinton, MA, USA) was used to detect the bioluminescence for mice injected with cancer cells, and the photon signal intensity (measured in total photons/s) was quantified for each region of interest using Living Image software version 2.50 (Caliper Life Sciences, Hopkinton, MA, USA). Imaging was performed every 2 min until the peak signal was observed (approximately 10–20 min post-injection).

### 2.7. Micro-Computed Tomography (µCT)

Hindlimbs (femur plus tibia) were excised from tumor-bearing mice, and soft tissues were removed without disrupting the tumors. Hindlimbs were fixed in formalin for 48 h, followed by storage in 70% ethanol. Hindlimbs were placed horizontally in a 36 mm diameter scanning holder and imaged using a 36 µm voxel size, 70 kVp, 114 µA with an integration time of 200 ms, and 250 projections per 180° for whole bone scans. Hindlimbs were then placed vertically in a 12 mm diameter scanning holder and imaged using a 12 µm voxel size, 70 kVp, 114 µA with an integration time of 200 ms, and 500 projections per 180° for morphometric evaluation of the metaphyseal region of the tibia. This was performed using the µCT40 desktop cone-beam µCT scanner (Scanco Medical AG, Brüttisellen, Switzerland) using µCT software v6.4-2). Scans were reconstructed into 2D slices, all slices were analyzed using the µCT Evaluation Program (v6.5-2, Scanco Medical), and 3D reconstruction was performed using µCT Ray v4.2. The volume and mineral density of the total bone, trabecular bones (456.5–1050 mg HA/ccm), cortical bones (>1000 mg HA/ccm), and new bone outside the cortex (not including cortex, 456.5–1050 mg HA/ccm) were quantified in a 2.5 mm volume of interest starting 1 mm below the proximal tibial physis.

### 2.8. Histopathology

Mouse hindlimbs from the intracardiac injection experiments were fixed in 10% neutral-buffered formalin for 5 days at 4 °C, decalcified with Formical-2000^TM^ decalcifier (Thermo Scientific) for 24 h at 4 °C, embedded in paraffin, cut in 4 µm sections, and then stained with hematoxylin and eosin (H&E). Mouse hindlimbs from the intratibial injection experiments were fixed in 10% neutral-buffered formalin for 24 h and then decalcified with 10% EDTA for 10 days at 4 °C. Decalcified bones were then processed, embedded in paraffin, and cut into 5 µm sections. Consecutive sections from each sample were stained with H&E or tartrate-resistant acid phosphatase (TRAP). The TRAP stain was performed by incubating bone sections in 0.2 M acetate buffer for 20 min and subsequently in the same buffer with naphthol AS-MX phosphate (0.5 mg/mL) and fast red TR salt (1.1 mg/mL) for 30–45 min in a 37 °C oven. The color change was monitored every 15 min until the TRAP-positive area turned red. TRAP-stained sections were then counterstained with hematoxylin. Histological images of the slides were scanned with a slide scanner (MoticEasyScan Pro 6). Bone histomorphometry was performed using ImageScope software (version 11.2; Leica Biosystems, Buffalo Grove, IL, USA). The area of interest was selected on each section starting 1 mm below the proximal tibia epiphysis to 1 mm above the distal tibia epiphysis. Within the selected frame, the areas of tumor, medullary woven bone, and periosteal new bone (not including cortex) were measured.

### 2.9. Protein Extraction and Immunoblots

Canine PCa cells were lysed using MER lysis buffer (Thermo Scientific) according to the manufacturer’s protocol. Total protein (10 μg) was separated by SDS-PAGE and transferred to PVDF membranes (Bio-Rad). Membranes were blocked with 5% non-fat milk, probed with primary antibodies for DKK-1 (#4687), non-phospho-β-Catenin (#4270), phospho-JNK, JNK (from antibody sampler kit #9912), cleaved caspase-9, caspase-9 (from antibody sampler kit #9915), phospho-p65, p65 (from antibody sampler kit #9936), and β-actin (#4967) (all the primary antibodies were purchased from Cell Signaling Technology, Danvers, MA, USA, and used in a 1:1000 dilution), followed by incubation with HRP-conjugated secondary antibodies (1:2000, Cell Signaling Technology), and then detected by chemiluminescence using Quantity One imaging software (version 4.6.7) on a Bio-Rad Gel Docking system.

### 2.10. Statistical Analysis

Data were displayed as mean ± standard deviation or mean + standard deviation. Statistical analyses on mice or cells between two comparison groups were performed using an unpaired Student’s *t*-test, and for experiments with more than two groups, one-way ANOVA with Tukey’s multiple comparisons test or two-way ANOVA with Sidak’s multiple comparisons test was used. Statistical data analysis was performed using GraphPad Prism version 6.03 (La Jolla, CA, USA). Data with *p* < 0.05 were considered statistically significant.

## 3. Results

### 3.1. DKK-1 Did Not Affect the Incidence of Cancer Metastases but Induced Larger Tumors at Metastatic Bone Sites

Probasco or Probasco + DKK-1 cells were injected into the left ventricle of nude mice and allowed to grow in vivo for 5 weeks. The incidence of cancer metastasis was measured by bioluminescent (BLI) imaging (Appendix A). All mice with successful intracardiac injections (IC) developed bone metastases. The locations of bone metastases were similar, with appendicular long bones being the most common site, followed by the mandible and vertebrae (Figure 1a and Table 2). The number of metastatic sites ranged from 4 to 7 (average 5.4) metastases per mouse with Probasco cells and 5–8 (average 6.5) metastases per mouse with Probasco + DKK-1 cells (Figure 1b). There was no significant difference in the incidence of cancer metastases between the two groups, although a slight increase in the total number of metastatic sites was seen in the Probasco + DKK-1 group. BLI intensity from all bone metastatic sites was quantified at week 5. Probasco + DKK-1 mice developed larger tumors at sites of bone metastasis compared to the Probasco group, with a significant difference observed in the femora/tibias (Figure 1c). Histological examinations were then performed on tibias. Tibias without metastatic tumor cells had an intact cortex and pre-existing trabecular bones (Figure 1d(i,ii)). Probasco cells that metastasized to the tibia developed smaller tumors containing pseudocystic cavities and induced woven bone lined by cuboidal osteoblasts and neoplastic cells within the medullary cavity (Figure 1d(iii,iv)). In contrast, Probasco + DKK-1 metastases developed larger tumors that grew in/on both the intramedullary cavity and periosteum, displacing bone marrow cells, and had less induction of woven bone formation in the metaphysis (Figure 1d(v,vi)).

### 3.2. DKK-1 Increased Cancer Growth and Decreased Cancer-Induced Bone Formation

To directly investigate the cancer–bone interactions, PCa cells were injected into tibias. Consistent with the results of IC injection, Probasco + DKK-1 cells grew faster in the tibia and had significantly larger tumors at weeks 3 and 4 compared to Probasco cells (Figure 2a,b). µCT (micro-Computed Tomography) reconstruction images of whole tibias and proximal tibial metaphysis regions demonstrated that Probasco + DKK-1 caused marked bone destruction. Probasco + DKK-1 induced extensive periosteal new bone formation with a palisading pattern of bone emanating from the cortical surface, while Probasco induced woven bone that was confined to the endosteal surface and marrow cavities (Figure 2c). µCT analysis showed that the DKK-1 mice had less cortical bone volume and increased periosteal new bone formation compared to the Probasco group (Figure 2d,e). Histological assessment, consistent with the findings of µCT, showed that Probasco metastases induced intramedullary woven bone (Figure 3a(i,ii),b). In contrast, there were few trabeculae inside the tibia in the DKK-1 group (Figure 3a(iii,iv),b). The DKK-1 metastases induced cortical bone resorption, which enabled cancer cells to penetrate the cortex and grow outside the bone, leading to new periosteal bone formation (Figure 3c). TRAP-stained sections revealed that osteoclasts were present on the surface of tumor-induced intramedullary woven bone in Probasco mice (Figure 3a(v,vi)), which indicated increased pathologic bone remodeling. In DKK-1 mice, osteoclasts were observed on both the intact and resorbed cortical endosteum and the surface of medullary woven bones (Figure 3a(vii,viii)). Additionally, some remaining osteoclasts were found within DKK-1 tumors that grew outside the bone. Together, these results demonstrated that DKK-1 increased PCa growth, decreased cancer-induced intramedullary woven bone formation, and caused dramatic bone destruction in vivo.

### 3.3. Effects of DKK-1 on Osteoblasts and Osteoclasts

To determine how Probasco + DKK-1 cells affect bone cells, primary osteoclasts and osteoblasts were treated with conditioned media (CM) collected from cancer cells. No difference was found in the mRNA expression of the osteoclast activity markers, *Mmp9* and *Ctsk*, between groups (Figure 4a, Appendix A). TRAP staining showed that the number of osteoclasts was similar between treated groups, and the multinucleated cells were slightly larger in Probasco + DKK-1 CM-treated osteoclasts (Figure 4b, Appendix A). The ratio of *Opg*/*Rankl*, which regulates osteoclast activity, was decreased in Probasco + DKK-1 CM-treated osteoblasts with an upregulation in mRNA expression of *Rankl* and no change in *Opg*—promoting osteoclast activity (Figure 4c). In addition, the mineralization of osteoblasts was significantly inhibited by Probasco + DKK-1 CM compared to Probasco CM as measured by von Kossa staining (Figure 4d). These data suggested that the reduced intramedullary bone formation in Probasco + DKK-1 bone metastases in vivo might be due to the inhibition of osteoblast differentiation and the activation of osteoclasts via a decreased OPG/RANKL ratio in the bone microenvironment.

### 3.4. DKK-1 Stimulated PCa Cell Proliferation and Migration and Induced EMT

Probasco and Probasco + DKK-1 cells grew in an anchorage-dependent manner in vitro and formed monolayer sheets of tightly packed polygonal cells. However, Probasco + DKK-1 cells were smaller with a mesenchymal cell morphology (Figure 5a) and had a significantly increased proliferation rate compared to Probasco cells (Figure 5b). In addition, DKK-1 stimulated cell migration with the wound healing assay showing that gaps were fully repaired by Probasco + DKK-1 cells at 12 h, while Probasco cells required 24 h (Figure 5c). Based on the difference in morphology and migration rate of the two cell lines, the expression of epithelial–mesenchymal transition (EMT)-related genes was investigated. As expected, the epithelial marker, *CDH1*, was decreased. For mesenchymal markers, *SNAIL* and *TWIST1* were significantly increased in Probasco + DKK-1 cells, and there was no change in *SLUG*. The high expression of *DKK-1* in the transfected Probasco + DKK1 cells was also confirmed (Figure 5d). These results indicate that DKK-1 had an autocrine effect on Probasco cells, which stimulated cell proliferation, enhanced cell migration, and induced EMT.

### 3.5. The Cancer-Promoting Effect of DKK-1 Was Independent of the Canonical WNT Pathway

To explore the mechanism of the cancer-promoting effect of DKK-1, several cancer-associated signaling pathways were investigated. We first confirmed the protein level of human DKK-1 in the transfected Probasco + DKK-1 cells (Figure 6a,b). Since DKK-1 is known as an inhibitor of canonical WNT signaling, we determined whether the level of activated β-catenin was suppressed. As shown by western blots, there was no difference in the expression of non-phospho-β-catenin (the active form of β-catenin) (Figure 6a,c), suggesting that DKK-1 did not inhibit canonical WNT signaling. Since it has been reported that DKK-1 stimulated tumor growth by switching the WNT pathway from canonical to non-canonical WNT/JNK signaling, we next determined whether JNK was upregulated in Probasco + DKK-1 cells. Unexpectedly, the level of phospho-JNK (the active form of JNK) was dramatically decreased in DKK-1 cells (Figure 6a,d), indicating an inhibition of the JNK downstream pathway.

Considering JNK is a hub transcription factor involved in both cell survival and apoptosis, the regulatory role of DKK-1 on apoptosis signaling was detected. The protein level of cleaved caspase-9, a caspase-dependent apoptosis inducer, was significantly decreased in Probasco + DKK-1 cells (Figure 6a,e), which suggested the downregulation of apoptosis in the DKK-1 cells. In addition, since the expression of NF-kB has been linked to malignancy, including PCa, and there is a reciprocal relation between NF-kB and JNK [22], the activation of NF-kB signaling was measured. The NF-kB transcription factor, p-p65 (active form), and downstream target genes, *COX2* and *VEGFA*, had increased expression in DKK-1 cells, which demonstrated that NF-kB signaling was upregulated in the Probasco + DKK-1 cells (Figure 6a,f–h). These results indicated that the cancer-promoting effect of DKK-1 might be associated with the downregulation of apoptosis by inhibiting JNK signaling and the stimulation of cell survival by activating NF-kB signaling.

## 4. Discussion

DKK-1 functions as a cancer-promoting factor in PCa. We previously reported that DKK-1 increased the growth and bone metastasis of the osteoblastic/osteolytic canine Ace-1 PCa cell line [13]. The data from our present study are consistent with our previous findings and results reported on the role of DKK-1 in the human C4-2B cells [27]. Here, we showed that DKK-1 had a similar role in the cancer progression of the osteoblastic canine Probasco cell line. In clinical specimens from patients with PCa, it was observed that the expression of DKK-1 was significantly increased in PCa tissue compared to benign tissue, and high DKK-1 serum levels were associated with shorter overall and disease-free survival [28]. Overexpression of DKK-1 has also been detected in breast, lung, and esophageal carcinomas, especially in patients presenting with bone metastases [29,30]. DKK-1 was found to exert cancer-promoting activity via its effects on tumor growth, metastasis, and angiogenesis [31]. Thus, DKK-1 may be an important marker for predicting cancer progression in patients with early malignancies.

Mechanisms of the cancer-promoting effect of DKK-1 vary based on the cancer context. Several studies have shown that DKK-1 contributed to cancer progression independent of the canonical WNT signaling. A study on osteosarcoma demonstrated that DKK-1 induced cancer stem cell-like properties by shifting the balance of WNT signaling in favor of the non-canonical WNT/JNK signaling [11]. In addition, the activation of the P13K/AKT pathway by the binding of DKK-1 to the novel CKAP4 receptor resulted in increased proliferation of normal canine kidney epithelial cells and human pancreatic and lung cancer cells [15]. Another study reported that DKK-1 binds CKAP4 to activate NF-kB signaling and drug resistance in multiple myeloma [32]. In our study, we found that the expression of JNK was dramatically decreased in Probasco + DKK-1 cells, leading to the downregulation of WNT/JNK signaling. This alternative outcome could be explained by the multifaceted role of JNK in cell proliferation and apoptosis or possibly the regulation of other pathways induced by DKK-1–CKAP4 interactions.

DKK-1 also decreased the osteoblastic phenotype of Probasco cells. DKK-1 has been proposed as a molecular switch that transitions the phenotype of PCa bone metastatic lesions from osteolytic to osteoblastic [33]. In clinical data, a progressive decline in DKK-1 from primary tumors to bone metastases has been observed, which can unmask the osteoblastic activity of WNTs in the bone microenvironment [34]. In experimental models, DKK-1 is naturally highly expressed in the human osteolytic PC3 cell line and nearly undetectable in the osteoblastic C4-2B cells [35]. Knock-down of *DKK-1* in PC3 converted it to an osteoblastic phenotype [27]. Conversely, overexpression of *DKK-1* in C4-2B decreased its osteoblastic activity [27]. In supporting these findings, we also showed that Probasco + DKK-1 tumors induced less medullary woven bone formation compared to Probasco tumors. PCa-induced woven bone formation was often found in the metaphyseal region of the medullary cavity, as seen in men with advanced PCa [36]. The periosteal new bone observed in the DKK-1 group was likely induced by the non-specific periosteal reaction [37] caused by cortical lysis rather than cancer-specific induction. In addition, decreased cortical bone volume in the DKK-1 group suggested that DKK-1 induced bone resorption, which facilitated the cancer cells to penetrate the cortical bone.

DKK-1 had little impact on the canonical WNT signaling in Probasco cells. The activation of canonical WNT signaling is due to the accumulation and translocation of non-phosphorylated β-catenin from the cytoplasm to the nucleus. The inhibitory effect of DKK-1 on WNT signaling results in the phosphorylation and degradation of β-catenin in the cytoplasm [38]. In contrast, here we found that non-phosphorylated β-catenin (the active form) was not decreased in Probasco + DKK-1 cells. In addition, we previously showed that there was a minimal difference in nuclear β-catenin between Ace-1 cells and Ace-1 cells expressing DKK-1 [18]. Similarly, an insignificant impact on canonical WNT was detected in breast cancer MDA-MB-231 cells and prostate cancer PC3 cells, which produce large amounts of DKK-1 [35]. One explanation was the resistance to the action of DKK-1 due to the downregulation of Kremen receptors, as proposed by Clines et al. [35]. DKK-1 exerts its inhibitory effect on canonical WNT signaling by binding to Kremen receptors and sequentially sequestering co-receptor LRP5/6 away from WNT ligands and frizzle receptors [38]. The study by Clines et al. showed that the expression of Kremen was decreased in cancer cells secreting high levels of DKK-1, resulting in no change in the activation of canonical WNT [35]. Future studies could be conducted to detect the expression of Kremen and LRP5/6 in canine PCa cells to investigate the resistance to DKK-1.

DKK-1 can be a potential therapeutic target to prevent PCa progression. Studies on the development of DKK-1 inhibitors have been carried out in recent years, and several strategies have produced encouraging clinical results in different pathological models [39]. DKN-01, an anti-DKK-1 antibody, has been tested in a PCa mouse xenograft model with elevated DKK-1, showing significant inhibitory effects on tumor growth and angiogenesis in a natural killer cell-dependent manner [40]. A clinical phase I/II trial (NCT03837353) has been conducted to investigate DKN-01 as monotherapy or combined with docetaxel for treating advanced PCa with high DKK-1 levels. In addition, a subtype of metastatic castration-resistant PCa termed double-negative PCa, characterized by low androgen receptors (AR) and no neuroendocrine signaling, has increased DKK-1 relative to prostate-specific antigen (PSA), suggesting DKK-1 as a possible biomarker and pharmaceutical target for a substantial portion of PCa with low AR and PSA [40]. Canine PCa lacks the expression of PSA and is usually androgen-independent [23,41,42,43], and the mRNA expression of *PSA* and *AR* in Probasco cells is not detectable [25]. Probasco + DKK-1 cells, with the characteristics of developing bone metastases, being AR-negative, and having high DKK-1 expression, could serve as a valuable model for investigating DKK-1-targeted treatments in metastatic castration-resistant PCa.

## 5. Conclusions

We have shown that over-expression of DKK-1 in the canine osteoblastic PCa cell line Probasco stimulated cancer growth and attenuated the osteoblastic phenotype of bone metastases. Our findings, together with previous reports, demonstrated the regulatory effects of DKK-1 in the cancer–bone microenvironment, supported multiple roles of DKK-1 in noncanonical WNT and other signaling pathways, and indicated that DKK-1 could be a promising therapeutic target for PCa.

## Figures and Tables

**Figure 1 cells-12-02695-f001:**
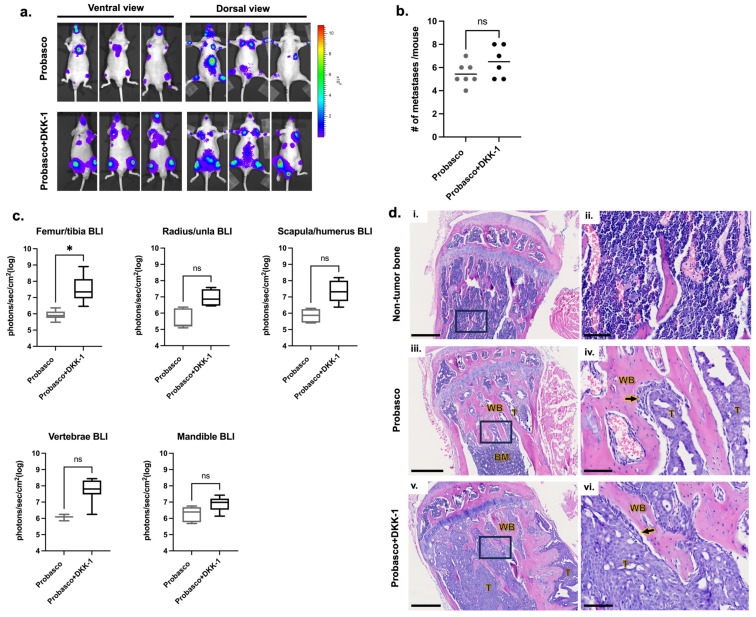
Probasco or Probasco + DKK-1 cells were injected into the left cardiac ventricle of athymic nude mice. Mice were euthanized at week 5. (**a**) Representative bioluminescent images of mice showing the metastatic sites at week 5. (**b**) The quantification of the total number of metastasis sites based on bioluminescent signals at week 5 (signals detected from the chest region were due to the growth of cancer cells that leaked during intracardiac injections and were not counted as metastatic sites). Data were displayed as scatter plots with mean lines. (**c**) Quantification of bioluminescent intensity from metastatic bone sites at week 5. Bioluminescent intensity is proportional to the number of viable tumor cells and reflects tumor size. Data were displayed as box and whisker plots. (**d**) H&E section of the non-tumor tibia (**i**,**ii**) and tumor-bearing tibia. Probasco formed smaller tumors and caused intramedullary woven bone formation (**iii**,**iv**). Probasco + DKK-1 had larger tumors that grew inside and outside the bone and induced predominant periosteal new bone formation (**v**,**vi**) (Enlarged images of black rectangular regions in (**i**,**iii**,**v**) were shown in (**ii**,**iv**,**vi**); T, tumor; BM, bone marrow; WB, tumor-induced woven trabecular bone; arrows, osteoblasts; bars, (**i**,**iii**,**v**) = 500 µm; (**ii**,**iv**,**vi**) = 100 µm). *n* = 7 for Probasco group, *n* = 6 for Probasco + DKK-1 group. Data were analyzed using an unpaired *t*-test. ns: no significance, * *p* < 0.05. For all figures, significant differences are marked with asterisks.

**Figure 2 cells-12-02695-f002:**
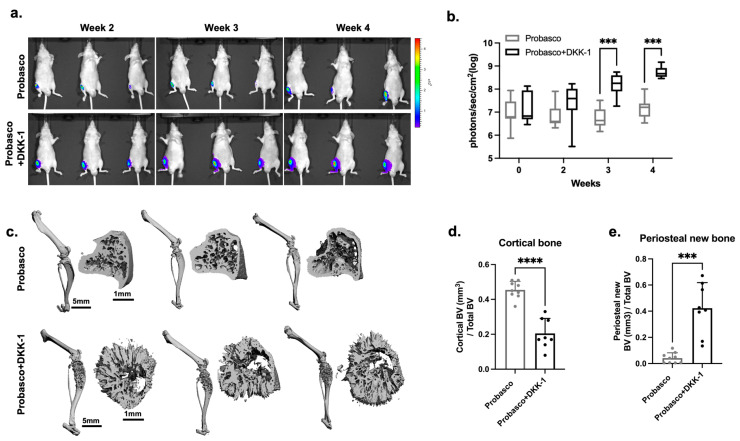
Probasco or Probasco + DKK-1 cells were injected into the proximal tibias of athymic nude mice (*n* = 9 for Probasco group, *n* = 8 for Probasco + DKK-1 group). Mice were euthanized at week 4. The growth of tumors was monitored by bioluminescence, and the structure of bones was assessed by µCT (micro-computed tomography). (**a**) Representative bioluminescent images of tumors at weeks 2, 3, and 4 post-injection. (**b**) Quantification of tumor growth based on bioluminescent intensity. Two-way ANOVA with Sidak’s multiple comparisons test, *** *p* < 0.001. Data were displayed as box and whisker plots. (**c**) Representative µCT-scanned images of the whole hindlimb and proximal tibial metaphysis regions (2.5 mm volume of interest starting 1 mm below the proximal tibial metaphysis) showed the 3D-reconstructed structure of tumor-bearing tibias. Quantification of µCT demonstrated the bone volume fractions of cortical bone (**d**) and periosteal new bone (**e**). Data were analyzed using an unpaired *t*-test. *** *p* < 0.001, **** *p* < 0.0001. Data were displayed as mean + SD.

**Figure 3 cells-12-02695-f003:**
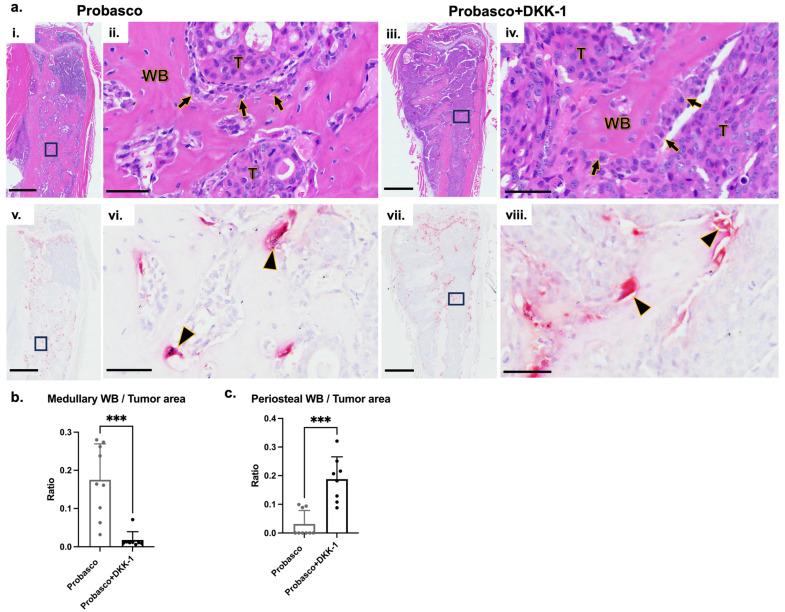
Histological assessment of intratibial-injected bones. (**a**) Representative low-magnification and high-magnification (rectangular regions) images of tibia bone sections stained with H&E (**i**–**iv**) and TRAP (**v**–**viii**) (Enlarged images of black rectangular regions in (**i**,**iii**,**v**,**vii**) were shown in (**ii**,**iv**,**vi**,**viii**); WB, tumor-induced woven bone; T, tumors; arrow, osteoblasts; arrowhead, osteoclasts; bars, (**i**,**iii**,**v**,**vii**) = 1 mm; (**ii**,**iv**,**vi**,**viii**) = 50 µm). (**b**) Quantitative histomorphometric analysis displayed the ratio of the intramedullary woven bone area to the tumor area. (**c**) The ratio of periosteal new bone area to the tumor area. *n* = 9 for Probasco group, *n* = 8 for Probasco + DKK-1 group. *t*-test was used for data analysis. *** *p* < 0.001. Data were displayed as mean + SD.

**Figure 4 cells-12-02695-f004:**
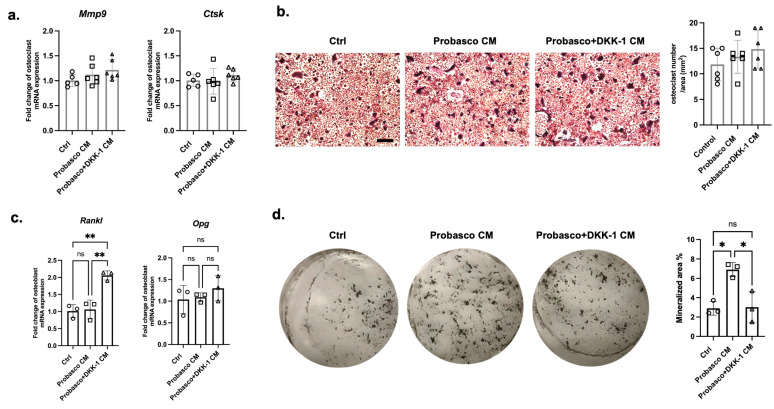
Effects of Probasco or Probasco + DKK-1 conditioned medium (CM) on primary murine osteoclasts and osteoblasts. (**a**) mRNA expression of *Mmp9* and *Ctsk* in control (ctrl) and CM-treated osteoclasts. (**b**) Representative images and quantification of TRAP-stained osteoclasts. TRAP-positive osteoclasts were stained with a red to purple color, and the number of multinucleated osteoclasts was quantified as osteoclast number per area (mm^2^). Bar = 200 µm. *n* = 5 to 6 for each osteoclast group. (**c**) mRNA expression of *Rankl* and *Opg* in osteoblasts. (**d**) von Kossa staining for the mineralization of osteoblast cultures, and the mineralized area was quantified. *n* = 3 for each osteoblast group. Data were analyzed using one-way ANOVA with Tukey’s multiple comparison test. ns: no significance, * *p* < 0.05, ** *p* < 0.01. Data were displayed as mean ± SD.

**Figure 5 cells-12-02695-f005:**
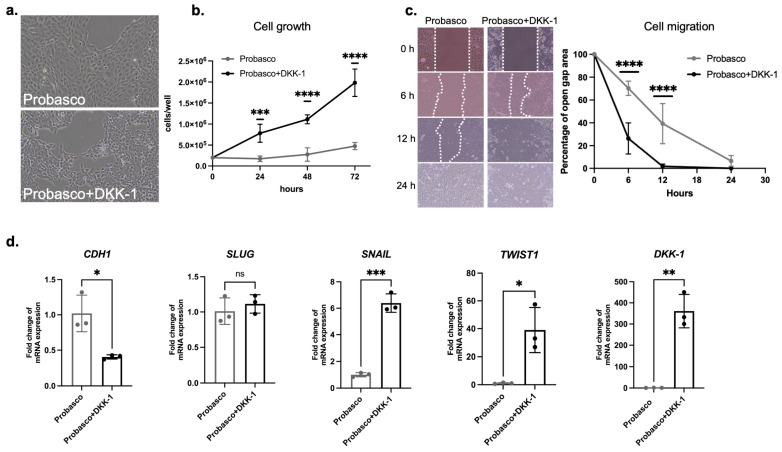
In vitro morphology, growth and migration, and gene expression of Probasco and Probasco + DKK-1 cells. (**a**) Phase contrast microscopy of cells in culture. (**b**) In vitro cell growth. n = 3 for each group, two-way ANOVA with Sidak’s multiple comparisons test, *** *p* < 0.001, **** *p* < 0.0001. (**c**) Representative images and quantification of in vitro cell migration. *n* = 4 for each group, two-way ANOVA with Sidak’s multiple comparisons test, **** *p* < 0.0001. (**d**) mRNA expression of human *DKK-1* and epithelial–mesenchymal transition (EMT)-related genes (*CDH1*, *SLUG*, *SNAIL*, *TWIST1*) in Probasco + DKK-1 cells compared to Probasco cells. *n* = 3 for each group, *t*-test, ns: no significance, * *p* < 0.05, ** *p* < 0.01, *** *p* < 0.001. Data were displayed as mean ± SD.

**Figure 6 cells-12-02695-f006:**
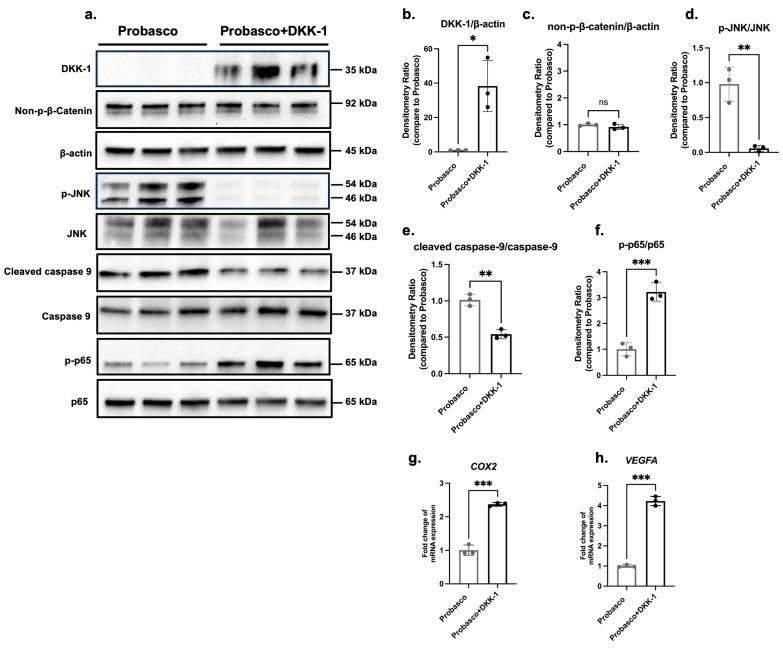
Alteration of signaling pathways by autocrine effects of DKK-1 in Probasco cells. (**a**) Western blots were performed on total protein lysates from cell lines for detecting (**b**) the overexpressed human DKK-1, (**c**) canonical WNT signaling; non-phospho β-catenin (active form) and β-actin (loading control), (**d**) non-canonical WNT/JNK signaling; phospho-JNK (active form) and total JNK, (**e**) apoptosis; cleaved caspase-9 (active form) and caspase-9, and (**f**) NF-kB/p65 signaling; phosphor p65 (active form) and total p65. Protein levels in DKK-1 cells were analyzed by densitometry and compared to levels in parental Probasco cells. (**g**,**h**) mRNA expression of NF-kB/p65 downstream target genes *COX2* and *VEGFA*. *n* = 3 for each group, *t*-test, ns: no significance, * *p* < 0.05, ** *p* < 0.01, *** *p* < 0.001. Data were displayed as mean ± SD.

**Table 1 cells-12-02695-t001:** Primer list.

Gene	Species	Forward Primer	Reverse Primer
*Ubc*	murine	CGTCGAGCCCAGTGTTACCACCAAGAAGG	CCCCCATCACACCCAAGAACAAGCACAAG
*Rankl*		ACACCTCACCATCAATGCT	CTTAACGTCATGTTAGAGATCTTGG
*Opg*		AGCTGCTGAAGCTGTGGAA	TCGAGTGGCCGAGAT
*Mmp9*		CATTCGCGTGGATAAGGAGT	ATTTTGGAAACTCACACGCC
*Ctsk*		CTTCCAATACGTGCAGCAGA	CCTCTGCATTTAGCTGCCTT
*GAPDH*	canine	CCCACTCTTCCACCTTCGAC	AGCCAAATTCATTGTCATACCAGG
*CDH1*		GCTGCTGACCTGCAAGGCGA	GGCCGGGGTATCGGGGACAT
*SLUG*		GGCAAGGCGTTTTCCAGACCCT	GGGCAAGAAAAAGGCTTCTCCCCAG
*SNAIL*		GTCTGTGGCACCTGCGGGAAG	GAAGGTTGGAGCGGTCGGCA
*TWIST1*		GGCAGGGCCGGAGACCTAGATG	TCCACGGGCCTGTCTCGCTT
*COX2*		AACATCCCCTTCCTGCGAAAT	TGGGTGTTGGACAGTCATCAG
*VEGFA*		CGCAGACGTGTAAATGTTCCTG	TCCCGAAACCCTGAGGGAG
*DKK-1*	human	GACCATTGACAACTACCAGCC	GGGACTAGCGCAGTACTCATC

**Table 2 cells-12-02695-t002:** Number of mice in each group that developed metastases at different bone sites.

	Femur/Tibia	Radius/Ulna	Scapula/Humerus	Vertebrae	Mandible
Probasco (*n* = 7)	7/7	3/7	3/7	2/7	6/7
Probasco + DKK-1 (*n* = 6)	6/6	5/6	5/6	4/6	6/6

## Data Availability

The data that support the findings of this study are available from the corresponding author upon reasonable request.

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
