# Peer review of "Effects of Dickkopf-1 (DKK-1) on Prostate Cancer Growth and Bone Metastasis"

_cells, 2023, doi:10.3390/cells12232695_

Round 1

Reviewer 1 Report

Comments and Suggestions for Authors

The authors investigated the role of DKK-1 on prostate cancer growth and bone metastasis, analysing also the alteration of bone remodeling activity. Yuan and co-authors performed in vivo and in vitro experiments to characterize the role of DKK-1. The papers is interesting but there are some issues that should be clarified:

1.       The authors described that the treatment of CM from Probasco+DKK1 culture does not affect MMP9 and CTSK expression of osteoclasts, but stimulates the formation of larger osteoclasts. The authors should evaluate the number of osteoclasts in cultures, the number of nuclei/osteoclast and expression of genes involved in osteoclast fusion including CD44, MFR and DC-Stamp.

2.       The effects of CM on osteoblasts should be further investigated, evaluating ALP activity of osteoblasts and gene expression analysis of Runx2, Osterix, Collagen and Osteocalcin.

3.       Change picture 4d with a better one.

4.       In vitro experiments of invasion should be performed.

Reviewer 2 Report

Comments and Suggestions for Authors

Wnt family proteins and DKK-1 are critical for regulating developmental and oncogenic processes. Given the in-depth understanding of the molecular mechanisms underlying the involvement of the Wnt signaling pathway in bone metastasis, new players to modulate the Wnt pathway for bone metastasis therapy would be exciting. 

Dickkopf-1 (Dkk-1) upregulates a noncanonical Wnt/JNK pathway, resulting in osteoclast stimulation, cell proliferation, and epithelial-to-mesenchymal transition (EMT) of cancer cells. DKK-1 has both autocrine and paracrine effects in the bone metastasis milieu, increasing tumor growth and decreasing the osteoblastic activity of prostate cancer (PCa). This study aimed to explore the molecular role of DKK-1 in PCa progression using the canine osteoblastic Probasco PCa cell line. The dog is the only species other than man to spontaneously develop PCa and subsequent osteoblastic bone metastases, making it valuable as a research model. 

PCa promotes osteoblast differentiation through canonical and noncanonical Wnt signaling pathways that stimulate BMP-dependent and independent osteoblast differentiation. DKK-1-mediated inhibition of osteoblasts, which contributes to tumor progression and osteolytic metastases, may also play a role in developing metastases with osteoblastic features. 

Intracardiac/IC injections lead to tumors in the bone, typical of breast and prostate cancer. Data from Fig.1 alone will suffice. The purpose of IT tumor instillation, however, is not clear. That being said, data from Fig. 2 is impressive. Similar mCT images are desirable for bone mets from Fig. 1.

Histomorphometry needs to be carried out at a specific and similar sites.

The conclusion drawn from Fig. 4 does not substantiate the intramedullary bone formation seen in vivo in Fig. 3. They had only obtained CM from the cells, and this experiment could not define the dynamic relationship between Wnts and BMPs in PCa cells.

Validation of DKK-1 overexpression is missing in Fig. 5/6. A simple WB to demonstrate the same should suffice.

Current progress in pharmacological Wnt modulators against cancer bone metastasis, discuss emerging therapeutic strategies based on Wnt pathway-related targets for bone therapy, and highlighting opportunities to harness the Wnt pathway for bone metastasis therapeutics are missing in the current version.

Recent studies on tumor–bone interactions presented multiple lines of evidence that support a tumor-suppressive role of Lrp5 in Wnt signaling. This concept has been downplayed in the manuscript.

Wnt signaling is crucial for the transmission of bone pain. Wnt5b and its co-receptor RYK had increased expression in the dorsal root ganglia of bone tumor-bearing mice. No reference to bone pain and Wnt had been discussed.

Round 2

Reviewer 1 Report

Comments and Suggestions for Authors

The authors should report in result section, that they did not find alterations of osteoclast number and they should insert the data of count. 

Author Response

We thank the reviewer's input to improve the manuscript. 

Quantification of osteoclast number has been added to Figure 4. The results part was updated accordingly. Please find the revisions highlighted in yellow in the resubmitted file. 

On behalf of all co-authors,

Shiyu Yuan

Reviewer 2 Report

Comments and Suggestions for Authors

Authors were responsive to previously raised concerns.

Author Response

We appreciate the reviewer's input and valuable suggestions to improve the quality of the manuscript.

On behalf of all co-authors,

Shiyu Yuan